# Drift of neural ensembles driven by slow fluctuations of intrinsic excitability

**Geoffroy Delamare[1]\*, Yosif Zaki[2], Denise J Cai[2], Claudia Clopath[1]\***

[1]Department of Bioengineering, Imperial College London, London, United Kingdom; [2]Department of Neuroscience, Icahn School of Medicine at Mount Sinai, New York, United States

**Abstract** Representational drift refers to the dynamic nature of neural representations in the brain despite the behavior being seemingly stable. Although drift has been observed in many different brain regions, the mechanisms underlying it are not known. Since intrinsic neural excitability is suggested to play a key role in regulating memory allocation, fluctuations of excitability could bias the reactivation of previously stored memory ensembles and therefore act as a motor for drift. Here, we propose a rate-based plastic recurrent neural network with slow fluctuations of intrinsic excitability. We first show that subsequent reactivations of a neural ensemble can lead to drift of this ensemble. The model predicts that drift is induced by co-activation of previously active neurons along with neurons with high excitability which leads to remodeling of the recurrent weights. Consistent with previous experimental works, the drifting ensemble is informative about its temporal history. Crucially, we show that the gradual nature of the drift is necessary for decoding temporal information from the activity of the ensemble. Finally, we show that the memory is preserved and can be decoded by an output neuron having plastic synapses with the main region.

**\*For correspondence:**
g.delamare21@imperial.ac.uk (GD);
c.clopath@imperial.ac.uk (CC)

## eLife assessment

This is an **important** theoretical study providing insight into how fluctuations in excitability can contribute to gradual changes in the mapping between population activity and stimulus, commonly referred to as representational drift. The authors provide **convincing** evidence that fluctuations can contribute to drift. Overall, this is a well-presented study that explores the question of how changes in intrinsic excitability can influence distinct memory representations.

## Introduction

In various brain regions, the neural code tends to be dynamic although behavioral outputs remain stable. Representational drift refers to the dynamic nature of internal representations as they have been observed in sensory cortical areas (*Driscoll et al., 2017*; *Sadeh and Clopath, 2022*; *Driscoll et al., 2022*) or the hippocampus (*Ziv et al., 2013*; *Hainmueller and Bartos, 2018*) despite stable behavior. It has even been suggested that pyramidal neurons from the CA1 and CA3 regions form dynamic rather than static memory engrams (*Hainmueller and Bartos, 2018*; *Spalla et al., 2021*), namely that the set of neurons encoding specific memories varies across days. In the amygdala, retraining of a fear memory task induces a turnover of the memory engram (*Cho et al., 2021*). Additionally, plasticity mechanisms have been proposed to compensate for drift and to provide a stable read-out of the neural code (*Rule and O'Leary, 2022*), suggesting that information is maintained. Altogether, this line of evidence suggests that drift might be a general mechanism with dynamical representations observed in various brain regions.

However, the mechanisms underlying the emergence of drift and its relevance for the neural computation are not known. Drift is often thought to arise from variability of internal states (*Sadeh and Clopath, 2022*), neurogenesis (*Rechavi et al., 2022*; *Driscoll et al., 2017*) or synaptic turnover (*Attardo et al., 2015*) combined with noise (*Kossio et al., 2021*; *Manz and Memmesheimer, 2023*). On the other hand, excitability might also play a role in memory allocation (*Zhou et al., 2009*; *Mau et al., 2020*; *Rogerson et al., 2014*; *Silva et al., 2009*), so that neurons having high excitability are preferentially allocated to memory ensembles (*Cai et al., 2016*; *Rashid et al., 2016*; *Silva et al., 2009*). Moreover, excitability is known to fluctuate over timescales from hours to days, in the amygdala (*Rashid et al., 2016*), the hippocampus (*Cai et al., 2016*; *Grosmark and Buzsáki, 2016*), or the cortex (*Huber et al., 2013*; *Levenstein et al., 2019*). Subsequent reactivations of a neural ensemble at different time points could therefore be biased by excitability (*Mau et al., 2022*), which varies at similar timescales than drift (*Mau et al., 2018*). Altogether, this evidence suggest that fluctuations of excitability could act as a cellular mechanism for drift (*Mau et al., 2020*).

In this short communication, we aimed at proposing how excitability could indeed induce a drift of neural ensembles at the mechanistic level. We simulated a recurrent neural network (*Delamare et al., 2022*) equipped with intrinsic neural excitability and Hebbian learning. As a proof of principle, we first show that slow fluctuations of excitability can induce neural ensembles to drift in the network. We then explore the functional implications of such drift. Consistent with previous works (*Rubin et al., 2015*; *Clopath et al., 2017*; *Mau et al., 2018*; *Miller et al., 2018*), we show that neural activity of the drifting ensemble is informative about the temporal structure of the memory. This suggest that fluctuations of excitability can be useful for time-stamping memories (*i.e.* for making the neural ensemble informative about the time at which it was form). Finally, we confirmed that the content of the memory itself can be steadily maintained using a read-out neuron and local plasticity rule, consistently with previous computational works (*Rule and O'Leary, 2022*). The goal of this study is to show one possible mechanistic implementation of how excitability can drive drift.

## Results

Many studies have shown that memories are encoded in sparse neural ensembles that are activated during learning and many of the same cells are reactivated during recall, underlying a stable neural representation (*Josselyn and Tonegawa, 2020*; *Poo et al., 2016*; *Mau et al., 2020*). After learning, subsequent reactivations of the ensemble can happen spontaneously during replay, retraining or during a memory recall task (e.g. following presentation of a cue *Josselyn and Tonegawa, 2020*; *Káli and Dayan, 2004*). Here, we directly tested the hypothesis that slow fluctuations of excitability can change the structure of a newly-formed neural ensemble, through subsequent reactivations of this ensemble.

To that end, we designed a rate-based, recurrent neural network, equipped with intrinsic neural excitability (Methods). We considered that the recurrent weights are all-to-all and plastic, following a Hebbian rule (Methods). The network was then stimulated following a 4day protocol: the first day corresponds to the initial encoding of a memory and the other days correspond to spontaneous or cue-induced reactivations of the neural ensemble (Methods). Finally, we considered that excitability of each neuron can vary on a day long timescale: each day, a different subset of neurons has increased excitability (*Figure 1a*, Methods).

### Fluctuations of intrinsic excitability induce drifting of neural ensembles

While stimulating the naive network on the first day, we observed the formation of a neural ensemble: some neurons gradually increase their firing rate (*Figure 1b and c*, neurons 10–20, time steps 1000–3000) during the stimulation. We observed that these neurons are highly recurrently connected (*Figure 1d*, leftmost matrix) suggesting that they form an assembly. This assembly is composed of neurons that have a high excitability (*Figure 1a*, neurons 10–20 have increase excitability) at the time of the stimulation. We then show that further stimulations of the network induce a remodeling of the synaptic weights. During the second stimulation for instance (*Figure 1b and c*, time steps 4000–6000), neurons from the previous assembly (10–20) are reactivated along with neurons having high excitability at the time of the second stimulation (*Figure 1a*, neurons 20–30). Moreover, across several days, recurrent weights from previous assemblies tend to decrease while others increase (*Figure 1d*).

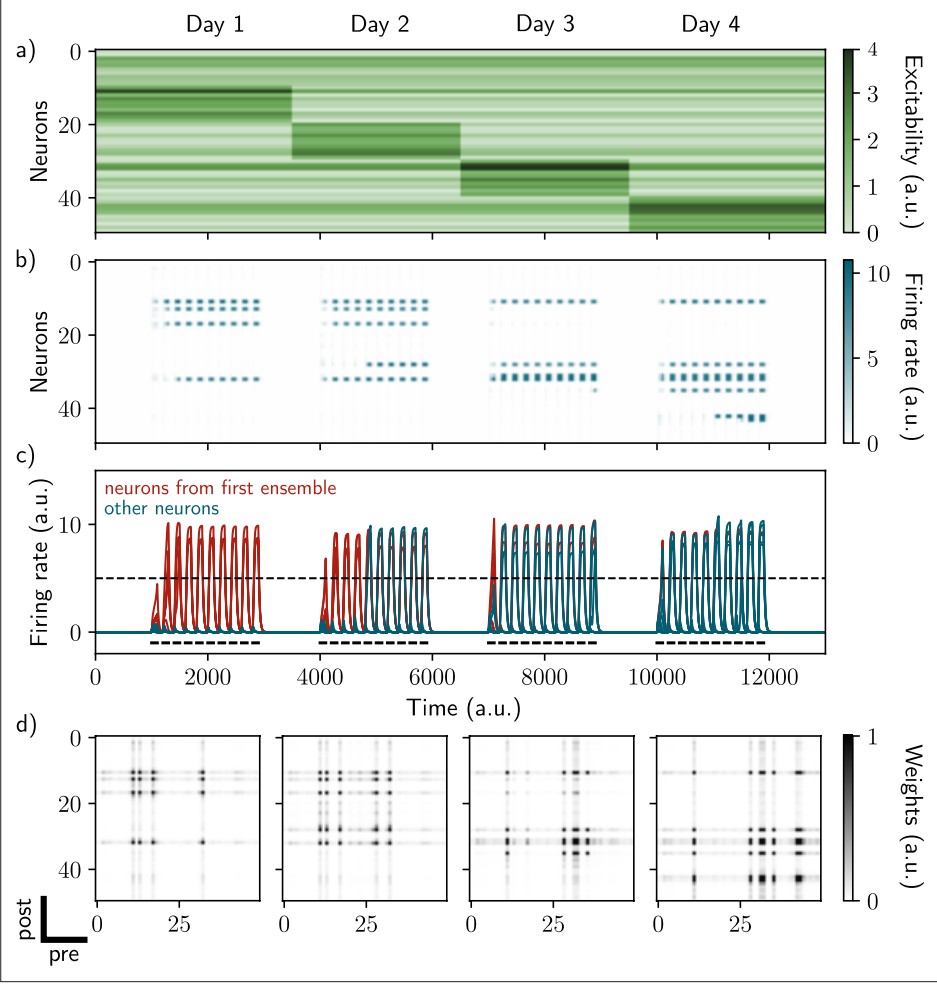

**Figure 1.** Excitability-induced drift of memory ensembles. (**a**) Distribution of excitability $\epsilon_i$ for each neuron $i$, fluctuating over time. During each stimulation, a different pool of neurons has a high excitability (Methods). (**b, c**) Firing rates of the neurons across time. The red traces in panel (**c**) correspond to neurons belonging to the first assembly, namely that have a firing rate higher than the active threshold after the first stimulation. The black bars show the stimulation and the dashed line shows the active threshold. (**d**) Recurrent weights matrices after each of the four stimuli show the drifting assembly.

The online version of this article includes the following figure supplement(s) for figure 1:

**Figure supplement 1.** Comparison of drifting behavior for different values of excitability amplitude.

**Figure supplement 2.** The rate of the drift does not depend on the size of the initial engram.

Indeed, neurons from the original assembly (*Figure 1c*, red traces) tend to be replaced by other neurons, either from the latest assembly or from the pool of neurons having high excitability. This is translated at the synaptic level, where weights from previous assemblies tend to decay and be replaced by new ones. Overall, each new stimulation updates the ensemble according to the current distribution of excitability, inducing a drift towards neurons with high excitability. Finally, in our model, the drift rate does not depend on the size of the original ensemble (*Figure 1—figure supplement 2*, Methods).

## Activity of the drifting ensemble is informative about the temporal structure of the past experience

After showing that fluctuations of excitability can induce a drift among neural ensembles, we tested whether the drifting ensemble could contain temporal information about its past experiences, as suggested in previous works (*Rubin et al., 2015*).

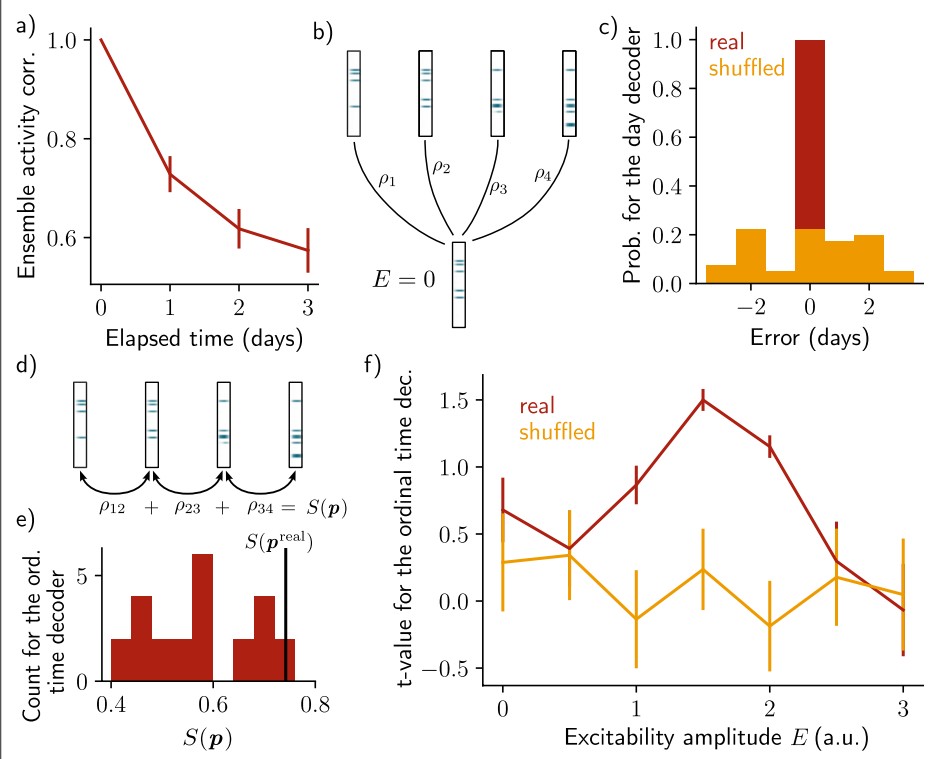

**Figure 2.** Neuronal activity is informative about the temporal structure of the reactivations. (**a**) Correlation of the patterns of activity between the first day and every other days, for n=10 simulations. Data are shown as mean ± s.e.m. (**b**) Schema of the day decoder. The day decoder maximises correlation between the patterns of each day with the pattern from the simulation with no increase in excitability. (**c**) Results of the day decoder for the real data (red) and the shuffled data (orange). Shuffled data consist of the same activity pattern for which the label of each cell for every seed has been shuffled randomly. For each simulation, the error is measured for each day as the difference between the decoded and the real day. Data are shown for n=10 simulations and for each of the 4 days. (**d**) Schema of the ordinal time decoder. This decoder output the permutation $\boldsymbol{p}$ that maximises the sum $S(\boldsymbol{p})$ of the correlations of the patterns for each pairs of days. (**e**) Distribution of the value $S(\boldsymbol{p})$ for each permutation of days $\boldsymbol{p}$. The value for the real permutation $S(\boldsymbol{p}^{\text{real}})$ is shown in black. (**f**) Student's test t-value for n=10 simulations, for the real (red) and shuffled (orange) data and for different amplitudes of excitability $E$. Data are shown as mean ± s.e.m. for n=10 simulations.

The online version of this article includes the following figure supplement(s) for figure 2:

**Figure supplement 1.** Sparse recurrent connectivity shows similar drifting behavior as all-to-all connectivity.

**Figure supplement 2.** Change of excitability as a variable slope of the input-output function shows similar drifting behavior as considering a change in the threshold.

**Figure supplement 3.** Two distinct ensembles can be encoded and drift independently.

**Figure supplement 4.** The two ensembles are informative about their temporal history and can be decoded using two output neurons.

Inspired by these works, we asked whether it was possible to decode relevant temporal information from the patterns of activity of the neural ensemble. We first observed that the correlation between patterns of activity after just after encoding decreases across days (***Figure 2a***, Methods), indicating that after each day, the newly formed ensemble resembles less the original one. Because the patterns of activity differ across days, they should be informative about the absolute day from which they were recorded. To test this hypothesis, we designed a day decoder (***Figure 2b***, Methods), following the work of ***Rubin et al., 2015***. This decoder aims at inferring the reactivation day of a given activity pattern by comparing the activity of this pattern during training and the activity just after memory encoding without increase in excitability (***Figure 2b***, Methods). We found that the day decoder perfectly outputs the reactivation day as compared to using shuffled data (***Figure 2c***).

After showing that the patterns of activity are informative about the reactivation day, we took a step further by asking whether the activity of the neurons is also informative about the order in which the memory evolved. To that end, we used an ordinal time decoder (Methods, as in **Rubin et al., 2015**) that uses the correlations between activity patterns for pairs of successive days, and for each possible permutation of days $p$ (**Figure 2d**, Methods). The sum of these correlations $S(p)$ differs from each permutation $p$ and we assumed that the neurons are informative about the order at which the reactivations of the ensemble happened if the permutation maximising $S(p)$ corresponds to the real permutation $p^{real}$ (**Figure 2e**, Methods). We found that $S(p^{real})$ was indeed statistically higher than $S(p)$ for the other permutations $p$ (**Figure 2f**, Student's t-test, Methods). However, this was only true when the amplitude of the fluctuations of excitability $E$ was in a certain range. Indeed, when the amplitude of the fluctuations is null, that is when excitability is not increased ($E = 0$), the ensemble does not drift (**Figure 1—figure supplement 1a**). In this case, the patterns of activity are not informative about the order of reactivations. On the other hand, if the excitability amplitude is too high ($E = 3$), each new ensemble is fully determined by the distribution of excitability, regardless of any previously formed ensemble (**Figure 1—figure supplement 1c**). In this regime, the patterns of activity are not informative about the order of the reactivations either. In the intermediate regime ($E = 1.5$), the decoder is able to correctly infer the order at which the reactivations happened, better than using the shuffled data (**Figure 2f**, **Figure 1—figure supplement 1b**).

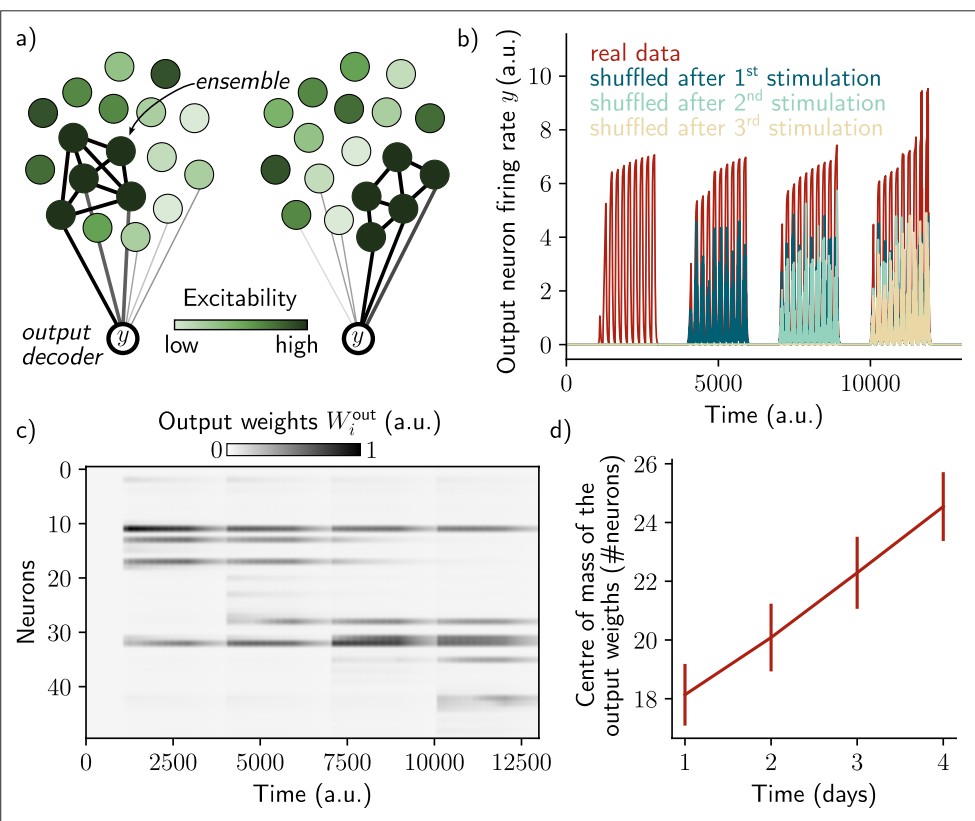

**Figure 3.** A single output neuron can track the memory ensemble through Hebbian plasticity. (**a**) Conceptual architecture of the network: the read-out neuron $y$ in red 'tracks' the ensemble by decreasing synapses linked to the previous ensemble and increasing new ones to linked to the new assembly. (**b**) Output neuron's firing rate across time. The red trace corresponds to the real output. The dark blue, light blue and yellow traces correspond to the cases where the output weights were randomly shuffled for every time points after presentation of the first, second and third stimulus, respectively. (**c**) Output weights for each neuron across time. (**d**) Center of mass of the distribution of the output weights (Methods) across days. The output weights are centered around the neurons that belong to the assembly at each day. Data are shown as mean ± s.e.m. for n=10 simulations.

The online version of this article includes the following figure supplement(s) for figure 3:

**Figure supplement 1.** The quality of the read-out decreases with the rate of the drift.

Finally, we sought to test whether the results are independent on the specific architecture of the model. To that end, we defined a change of excitability as a change in the slope of the activation function, rather than of the threshold (*Figure 2—figure supplement 2*, Methods). We also used sparse recurrent synaptic weights instead of the original all-to-all connectivity matrix (*Figure 2—figure supplement 1*, Methods). In both cases, we observed a similar drifting behavior and were able to decode the temporal order in which the memory evolved.

## A read-out neuron can track the drifting ensemble

So far, we showed that the drifting ensemble contains information about its history, namely about the days and the order at which the subsequent reactivations of the memory happened.

However, we have not shown that we could use the neural ensemble to actually decode the memory itself, in addition to its temporal structure. To that end, we introduced a decoding output neuron connected to the recurrent neural network, with plastic weights following a Hebbian rule (Methods). As shown by *Rule and O'Leary, 2022*, the goal was to make sure that the output neuron can track the ensemble even if it is drifting. This can be down by constantly decreasing weights from neurons that are no longer in the ensemble and increasing those associated with neurons joining the ensemble (*Figure 3a*). We found that the output neuron could steadily decode the memory (*i.e.* it has a higher firing than in the case where the output weights are randomly shuffled; *Figure 3b*). This is due to the fact that weights are plastic under Hebbian learning, as shown by *Rule and O'Leary, 2022*. We confirmed that this was induced by a change in the output weights across time (*Figure 3c*). In particular, the weights from neurons that no longer belong to the ensemble are decreased while weights from newly recruited neurons are increased, so that the center of mass of the weights distribution drifts across time (*Figure 3d*). Finally, we found that the quality of the read-out decreases with the rate of the drift (*Figure 3—figure supplement 1*, Methods).

## Two memories drift independently

Finally, we tested whether the network is able to encode two different memories and whether excitability could make two ensembles drift. On each day, we stimulated a random half of the neurons (context A) and the other half (context B) sequentially (Methods). We found that, day after day, the two ensembles show a similar drift than when only one ensemble was formed (*Figure 2—figure supplement 3*). In particular, the correlation between the patterns activity on the first day and the other days decay in a similar way (*Figure 2—figure supplement 4a*). For both contexts, the temporal order of the reactivations can be decoded for a certain range of excitability amplitude (*Figure 2—figure supplement 4b*). Finally, we found that using two output decoders allowed us to decode both memories independently. The output weights associated to both ensembles are remodeled to follow the drifting ensembles, but are not affected by the reactivation of the other ensemble (*Figure 2—figure supplement 4c*). Indeed, both neurons are able to 'track' the reactivation of their associated ensemble while not responding to the other ensemble (*Figure 2—figure supplement 4d*).

## Discussion

Overall, our model suggests a potential cellular mechanisms for the emergence of drift that can serve a computational purpose by 'time-stamping' memories while still being able to decode the memory across time. Although the high performance of the day decoder was expected, the performance of the ordinal time decoder is not trivial. Indeed, the patterns of activity of each day are informative about the distribution of excitability and therefore about the day at which the reactivation happened. However, the ability for the neural ensemble to encode the order of past reactivations requires drift to be gradual (*i.e.* requires consecutive patterns of activity to remain correlated across days). Indeed, if the amplitude of excitability is too low ($E = 0$) or too high ($E = 3$), it is not possible to decode the order at which the successive reactivations happened. This result is consistent with the previous works showing gradual change in neural representations, that allows for decoding temporal information of the ensemble (*Rubin et al., 2015*). Moreover, such gradual drifts could support complex cognitive mechanisms like mental time-travel during memory recall (*Rubin et al., 2015*).

In our model, drift is induced by co-activation of the previously formed ensemble and neurons with high excitability at the time of the reactivation. The pool of neurons having high excitability can

therefore 'time-stamps' memory ensembles by biasing allocation of these ensembles (*Clopath et al., 2017*; *Mau et al., 2018*; *Rubin et al., 2015*). We suggest that such time-stamping mechanism could also help link memories that are temporally close and dissociate those which are spaced by longer time (*Driscoll et al., 2022*; *Mau et al., 2020*; *Aimone et al., 2006*). Indeed, the pool of neurons with high excitability varies across time so that any new memory ensemble is allocated to neurons which are shared with other ensembles formed around the same time. This mechanism could be complementary to the learning-induced increase in excitability observed in amygdala (*Rashid et al., 2016*), hippocampal CA1 (*Cai et al., 2016*) and dentate gyrus (*Pignatelli et al., 2019*).

Finally, we intended to model drift in the firing rates, as opposed to a drift in the turning curve, of the neurons. Recent studies suggest that drifts in the mean firing rate and tuning curve arise from two different mechanisms (*Geva et al., 2023*; *Khatib et al., 2023*). Experience drives a drift in neurons turning curve while the passage of time drives a drift in neurons firing rate. In this sense, our study is consistent with these findings by providing a possible mechanism for a drift in the mean firing rates of the neurons driven a dynamical excitability. Our work suggests that drift can depend on any experience having an impact on excitability dynamics such as exercise as previously shown experimentally (*Rechavi et al., 2022*; *de Snoo et al., 2023*) but also neurogenesis (*Aimone et al., 2006*; *Tran et al., 2022*; *Rechavi et al., 2022*), sleep (*Levenstein et al., 2017*) or increase in dopamine level (*Chowdhury et al., 2022*).

Overall, our work is a proof of principle which highlights the importance of considering excitability when studying drift, although further work would be needed to test this link experimentally.

## Methods
### Recurrent neural network with excitability
Our rate-based model consists of a single region of $N$ neurons (with firing rate $r_i$, $1 \leq i \leq N$). All-to-all recurrent connections $W$ are plastic and follow a Hebbian rule given by:

$$\frac{dW_{ij}}{dt} = r_i * r_j / \tau_W - W_{ij} / \tau_{\text{decay}} \tag{1}$$

where $i$ and $j$ correspond to the pre- and post-synaptic neuron respectively. $\tau_W$ and $\tau_{\text{decay}}$ are the learning and the decay time constants of the weights, respectively.

A hard bound of $[0, c]$ was applied to these weights. We also introduced a global inhibition term dependent on the activity of the neurons:

$$I = I_0 + I_1 \sum_{i=1}^{N} r_i + I_2 \sum_{i=1}^{N} r_i^2 \tag{2}$$

here $I_0$, $I_1$ and $I_2$ are positive constants. All neurons receive the same input, $\Delta(t)$, during stimulation of the network (*Figure 1c*, black bars). Finally, excitability is modeled as a time-varying threshold $\epsilon_i$ of the input-output function of each neuron $i$. The rate dynamics of a neuron $i$ is given by:

$$\tau_r \frac{dr_i}{dt} + r_i = \text{ReLU} \left( \Delta(t) + \sum_{j=1}^{N} W_{ij} r_j - I + \epsilon_i(t) \right) \tag{3}$$

where $\tau_r$ is the decay time of the rates and ReLU is the rectified linear activation function. We considered that a neurons is active when its firing rate reaches the active threshold $\theta$.

In *Figure 2—figure supplement 1*, we applied a random binary mask to the recurrent weights in order to set 50% of the synapses at 0. A new mask was randomly sampled for each simulation.

In *Figure 2—figure supplement 2*, we modeled excitability as a change of the slope of the activation function (ReLU) instead of a change of the threshold as previously used (*Figure 2—figure supplement 2a*):

$$\tau_r \frac{dr_i}{dt} + r_i = \epsilon_i(t) * \text{ReLU} \left( \Delta(t) + \sum_{j=1}^{N} W_{ij} r_j - I \right) \tag{4}$$

## Protocol

We designed a 4-day protocol, corresponding to the initial encoding of a memory (first day) and subsequent random or cue-induced reactivations of the ensemble (*Josselyn and Tonegawa, 2020*; *Káli and Dayan, 2004*) (second, third, and fourth day). Each stimulation consists of $N_{\text{rep}}$ repetitions of interval $T$ spaced by a inter-repetition delay $IR$. $\Delta(t)$ takes the value $\delta$ during these repetitions and is set to 0 otherwise. The stimulation is repeated four times, modeling four days of reactivation, spaced by an inter-day delay $ID$. Excitability $\epsilon_i$ of each neuron $i$ is sampled from the absolute value of a normal distribution of mean 0 and standard deviation 1. In *Figure 2—figure supplement 2*, excitability $\epsilon_i$ is sampled from the absolute value of a normal distribution of mean 0.4 and standard deviation 0.2. Neurons 10–20, 20–30, 30–40, and 40–50 then receive an increase of excitability of amplitude $E$, respectively on days 1, 2, 3, and 4 (*Figure 1a*). A different random seed is used for each repetition of the simulation. When two memories were modeled (*Figure 2—figure supplements 3 and 4*), we stimulated a random half of the neurons (context A) and the other half (context B) successively (*Figure 2—figure supplement 3a*), every day.

## Decoders

For each day $d$, we recorded the activity pattern $V_d$, which is a vector composed of the firing rate of the neurons at the end of the last repetition of stimulation. To test the decoder, we also stimulated the network while setting the excitability at baseline ($E = 0$), and recorded the resulted pattern of activity $V_d^0$ for each day $d$. We then designed two types of decoders, inspired by previous works (*Rubin et al., 2015*): (1) a day decoder which infers the day at which each stimulation happened and (2) an ordinal time decoder which infers the order at which the reactivations occurred. For both decoders, the shuffled data was obtained by randomly shuffling the day label of each neuron. When two memories were modeled (*Figure 2—figure supplements 3 and 4*), the patterns of activity were taken at the end of the stimulations by context A and B, and the decoders were used independently on each memory.

1. The day decoder aims at inferring the day at which a specific pattern of activity occurred. To that end, we computed the Pearson correlation between the pattern with no excitability $V_d^0$ of the day $d$ and the patterns of all days $d'$ from the first simulation $V_{d'}$. Then, the decoder outputs the day $d_{\text{inf}}$ that maximises the correlation:

$$d_{\text{inf}} = \arg \max_{d'} \{ \text{corr}(V_d^0, V_{d'}) \} \tag{5}$$

The error was defined as the difference between the inferred and the real day $d_{\text{inf}} - d$.

2. The ordinal time decoder aims at inferring the order at which the reactivations happened from the patterns of activity $V_d$ of every day $d$. To that end, we computed the pairwise correlations of each pair of consecutive days, for the 4! possible permutations of days $\boldsymbol{p}$. The real permutation is called $\boldsymbol{p}^{\text{real}} = (1, 2, 3, 4)$ and corresponds to the real order of reactivations: day 1 → day 2 → day 3 → day 4. The sum of these correlations over the 3 pairs of consecutive days is expressed as:

$$S(\boldsymbol{p}) = \sum_{i=1}^{3} \text{corr}(V_{p_i}, V_{p_{i+1}}) \tag{6}$$

We then compared the distribution of these quantities for each permutation $\boldsymbol{p}$ to that of the real permutation $S(\boldsymbol{p}^{\text{real}})$ (*Figure 2*). The patterns of activity are informative about the order of reactivations if $S(\boldsymbol{p}^{\text{real}})$ corresponds to the maximal value of $S(\boldsymbol{p})$. To compare $S(\boldsymbol{p}^{\text{real}})$ with the distribution $S(\boldsymbol{p})$, we performed a Student's t-test, where the t-value is defined as:

$$t = \frac{S(\boldsymbol{p}^{\text{real}}) - \mu}{\sigma / \sqrt{N}} \tag{7}$$

where μ and σ correspond to the mean and standard deviation of the distribution $S(\boldsymbol{p})$, respectively.

The drift rate $\Delta$ (*Figure 1—figure supplement 2* and *Figure 3—figure supplement 1*) was computed as:

$$\Delta = \sum_{i=2}^{4} (1 - \text{corr}(V_1, V_i)) \tag{8}$$

## Memory read-out

To test if the network is able to decode the memory at any time point, we introduced a read-out neuron with plastic synapses to neurons from the recurrent network, inspired by previous computational works (*Rule and O'Leary, 2022*). The weights of these synapses are named $\mathbf{W}^{\text{out}} = (W_i^{\text{out}})_{1 \leq i \leq N}$ and follow the Hebbian rule defined as:

$$\frac{dW_{ij}^{\text{out}}}{dt} = h(\mathbf{W}^{\text{out}}) * r_i * y/\tau_{\text{out}}^+ - W_i^{\text{out}}/\tau_{\text{out}}^- \tag{9}$$

where $\tau_{\text{out}}^+$ and $\tau_{\text{out}}^-$ corresponds to the learning time and decay time constant, respectively. $h(\mathbf{W}^{\text{out}})$ is a homeostatic term defined as $h(\mathbf{W}^{\text{out}}) = 1 - \sum_{j=1}^{N} W_j^{\text{out}}$ which decreases to 0 throughout learning. $h$ takes the value 1 before learning and 0 when the sum of the weights reaches the value 1. $y$ is the firing rate of the output neuron defined $y$ as:

$$y = \sum_{i=1}^{N} W_i^{\text{out}} r_i \tag{10}$$

The read-out quality index $Q$ (*Figure 3—figure supplement 1*) was defined as:

$$Q = \langle \sum_{d=2}^{4} y_d / y_d^{\text{shuffle}} \rangle_{N_{\text{shuffle}}} \tag{11}$$

where $y_d$ corresponds to the value of $y$ taken at the end of the last repetition of day $d$, and $y_d^{\text{shuffle}}$ the equivalent with shuffled outputs weights. $\langle ... \rangle_{N_{\text{shuffle}}}$ indicates the average over $N_{\text{shuffle}} = 10$ simulations.

In *Figure 2—figure supplements 3 and 4*, two output decoders $y_k$, $k \in \{1, 2\}$, with corresponding weights $\mathbf{W}_k^{\text{out}} = (W_{i,k}^{\text{out}})_{1 \leq i \leq N}$ are defined as:

$$y_k = \sum_{i=1}^{N} W_{i,k}^{\text{out}} r_i + \beta_k \tag{12}$$

and follow the Hebbian rule defined as:

$$\frac{dW_{i,k}^{\text{out}}}{dt} = h(W_{i,k}^{\text{out}}) * r_i * y_k/\tau_{\text{out}}^+ - W_{i,k}^{\text{out}}/\tau_{\text{out}}^- \tag{13}$$

Then, we aimed at allocating $y_1$ and $y_2$ to the first and the second ensemble (context A and B), respectively. To that end, we used supervised learning on the first day by adding a current $\beta_k$ to the output neurons which is positive when the corresponding context is on:

$$\begin{aligned} \beta_0 &= 0.1 \text{ if } 1000 < t < 3000, 0 \text{ otherwise} \\ \beta_1 &= 0.1 \text{ if } 4000 < t < 6000, 0 \text{ otherwise} \end{aligned} \tag{14}$$

The shuffled traces were obtained by randomly shuffling the output weights $\mathbf{W}^{\text{out}}$ or $\mathbf{W}_k^{\text{out}}$ for each ensemble $k$.

## Table of parameters

The following parameters have been used for the simulations. When unspecified, the defaults values were used. All except $N$ are in arbitrary unit. *Figure 2—figure supplement 2* corresponds to the change from a threshold-based to a slope-based excitability. *Figure 2—figure supplements 3 and 4* corresponds to the stimulation of two ensembles. *Figure 2—figure supplement 1* corresponds to the sparsity simulation.

| Param. | Description | Default | Figure 2—figure supplement 2 | Figure 2—figure supplements 3 and 4 | Figure 2—figure supplement 1 |
|--------|-------------|---------|------------------------------|-------------------------------------|------------------------------|
| N | Number of neurons | 50 | - | - | - |

*Continued on next page*

*Continued*

| Param. | Description | Default | Figure 2—figure supplement 2 | Figure 2—figure supplements 3 and 4 | Figure 2—figure supplement 1 |
|---|---|---|---|---|---|
| $\tau_W$ | Learning time constant of the recurrent weights | 800 | 700 | - | - |
| $\tau_{decay}$ | Decay time constant of the recurrent weights | 1000 | 800 | 4000 | - |
| $\tau_r$ | Decay time constant of the firing rates | 20 | - | - | - |
| $\tau_{out}^+$ | Learning time constant of the output weights | 200 | - | - | - |
| $\tau_{out}^-$ | Decay time constant of the output weights | 1000 | - | - | - |
| $I_0$ | First inhibition parameter | 12 | 4 | 8 | 7 |
| $I_1$ | Second inhibition parameter | 0.5 | 0.7 | 0.8 | 0.8 |
| $I_2$ | Third inhibition parameter | 0.05 | - | - | - |
| $\delta$ | Input current during stimulation | 15 | - | 12 | 20 |
| $E$ | Amplitude of the fluctuations of excitability | 1.5 | 0.5 | - | - |
| $N_{rep}$ | Number of repetitions | 10 | - | - | - |
| $T$ | Duration of each repetition | 100 | - | - | - |
| $IR$ | Inter-repetition delay | 100 | - | - | - |
| $ID$ | Inter-stimulation delay | 1000 | - | - | - |
| $\theta$ | Active threshold | 5 | 1 | - | - |
| $c$ | Cap on the recurrent weights | 1 | .5 | - | - |

# Additional information

## Competing interests

Denise J Cai: Reviewing editor, *eLife*. The other authors declare that no competing interests exist.

## Funding

| Funder | Grant reference number | Author |
|---|---|---|
| Biotechnology and Biological Sciences Research Council | BB/N013956/1 | Claudia Clopath |
| Wellcome Trust | 10.35802/200790 | Claudia Clopath |
| Simons Foundation | 564408 | Claudia Clopath |
| Engineering and Physical Sciences Research Council | EP/R035806/1 | Claudia Clopath |

The funders had no role in study design, data collection and interpretation, or the decision to submit the work for publication. For the purpose of Open Access, the authors have applied a CC BY public copyright license to any Author Accepted Manuscript version arising from this submission.

## Author contributions

Geoffroy Delamare, Conceptualization, Data curation, Software, Formal analysis, Visualization, Methodology, Writing - original draft, Writing - review and editing; Yosif Zaki, Conceptualization, Writing - review and editing; Denise J Cai, Conceptualization, Project administration, Writing - review and

editing; Claudia Clopath, Conceptualization, Resources, Supervision, Funding acquisition, Validation, Investigation, Methodology, Project administration, Writing - review and editing

## Author ORCIDs
Geoffroy Delamare (iD) http://orcid.org/0000-0001-6217-4370
Denise J Cai (iD) http://orcid.org/0000-0002-7729-0523
Claudia Clopath (iD) http://orcid.org/0000-0003-4507-8648

Reviewer #3 (Public Review): https://doi.org/10.7554/eLife.88053.3.sa1
Author response https://doi.org/10.7554/eLife.88053.3.sa2

---

## Additional files

### Supplementary files
• MDAR checklist

### Data availability
The code for simulations and figures is available at GitHub (copy archived at *Delamare, 2024*).

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
