## [Editor Report · eLife assessment]

This is an **important** theoretical study providing insight into how fluctuations in excitability can contribute to gradual changes in the mapping between population activity and stimulus, commonly referred to as representational drift. The authors provide **convincing** evidence that fluctuations can contribute to drift. Overall, this is a well-presented study that explores the question of how changes in intrinsic excitability can influence distinct memory representations.

---

## [Referee Report · Reviewer #3 (Public Review)]

Summary of the findings:

The authors explore an important question concerning the underlying mechanism of representational drift, which despite intense recent interest remains obscure. The paper explores the intriguing hypothesis that drift may reflect changes in the intrinsic excitability of neurons. The authors set out to provide theoretical insight into this potential mechanism.

They construct a rate model with all-to-all recurrent connectivity, in which recurrent synapses are governed by a standard Hebbian plasticity rule. This network receives a global input, constant across all neurons, which can be varied with time. Each neuron also is driven by an "intrinsic excitability" bias term, which does vary across cells. The authors study how activity in the network evolves as this intrinsic excitability term is changed.

They find that after initial stimulation of the network, those neurons where the excitability term is set high become more strongly connected and are in turn more responsive to the input. Each day the subset of neurons with high intrinsic excitability is changed, and the network's recurrent synaptic connectivity and responsiveness gradually shift, such that the new high intrinsic excitability subset becomes both more strongly activated by the global input and also more strongly recurrently connected. These changes result in drift, reflected by a gradual decrease across time in the correlation of the neuronal population vector response to the stimulus.

The authors are able to build a classifier that decodes the "day" (i.e. which subset of neurons had high intrinsic excitability) with perfect accuracy. This is despite the fact that the excitability bias during decoding is set to 0 for all neurons, and so the decoder is really detecting those neurons with strong recurrent connectivity, and in turn strong responses to the input. The authors show that it is also possible to decode the order in which different subsets of neurons were given high intrinsic excitability on previous "days". This second result depends on the extent by which intrinsic excitability was increased: if the increase in intrinsic excitability was either too high or too low, it was not possible to read out any information about the past ordering of excitability changes.

Finally, using another Hebbian learning rule, the authors show that an output neuron, whose activity is a weighted sum of the activity of all neurons in the network, is able to read out the activity of the network. What this means specifically, is that although the set of neurons most active in the network changes, the output neuron always maintains a higher firing rate than a neuron with randomly shuffled synaptic weights, because the output neuron continuously updates its weights to sample from the highly active population at any given moment. Thus, the output neuron can read out a stable memory despite drift.

Strengths:

The authors are clear in their description of the network they construct and in their results. They convincingly show that when they change their "intrinsic excitability term", upon stimulation, the Hebbian synapses in their network gradually evolve, and the combined synaptic connectivity and altered excitability result in drifting patterns of activity in response to an unchanging input (Fig. 1, Fig. 2a). Furthermore, their classification analyses (Fig. 2) show that information is preserved in the network, and their readout neuron successfully tracks the active cells (Fig. 3). Finally, the observation that only a specific range of excitability bias values permits decoding of the temporal structure of the history of intrinsic excitability (Fig. 2f and Figure S1) is interesting, and as the authors point out, not trivial.

Weaknesses:

1. The way the network is constructed, there is no formal difference between what the authors call "input", Δ(t), and what they call "intrinsic excitability" Ɛ_i(t) (see Equation 3). These are two separate terms that are summed (Eq. 3) to define the rate dynamics of the network. The authors could have switched the names of these terms: Δ(t) could have been considered a global "intrinsic excitability term" that varied with time and Ɛ_i(t) could have been the external input received by each neuron in the network. In that case, the paper would have considered the consequence of "slow fluctuations of external input" rather than "slow fluctuations of intrinsic excitability", but the results would have been the same. The difference is therefore semantic. The consequence is that this paper is not necessarily about "intrinsic excitability", rather it considers how a Hebbian network responds to changes in excitatory drive, regardless of whether those drives are labeled "input" or "intrinsic excitability".

A revised version of the manuscript models "slope-based" excitability changes in addition to "threshold-based" changes. This serves to address the above concern that as constructed here changes in excitability threshold are not distinguishable from changes in input. However, it remains unclear what the model would do should only a subset of neurons receive a given, fixed input. In that case, are excitability changes sufficient to induce drift? This remains an important question that is not addressed by the paper in its current form.

1. Given how the learning rule that defines the input to the readout neuron is constructed, it is trivial that this unit responds to the most active neurons in the network, more so than a neuron assigned random weights. What would happen if the network included more than one "memory"? Would it be possible to construct a readout neuron that could classify two distinct patterns? Along these lines, what if there were multiple, distinct stimuli used to drive this network, rather than the global input the authors employ here? Does the system, as constructed, have the capacity to provide two distinct patterns of activity in response to two distinct inputs?

A revised version of the manuscript addresses this question, demonstrating that the network is capable of maintaining two distinct memories.

Impact:

Defining the potential role of changes in intrinsic excitability in drift is fundamental. Thus, this paper represents an important contribution. What we see here is that changes in intrinsic excitability are sufficient to induce drift. This raises the question for future work of the specific contributions of changing excitability from changing input to representational drift.

---

## [Author Response]

The following is the authors’ response to the latest reviews.

A revised version of the manuscript models "slope-based" excitability changes in addition to "threshold-based" changes. This serves to address the above concern that as constructed here changes in excitability threshold are not distinguishable from changes in input. However, it remains unclear what the model would do should only a subset of neurons receive a given, fixed input. In that case, are excitability changes sufficient to induce drift? This remains an important question that is not addressed by the paper in its current form.

Thank you for this important point. In the simulation of two memories (Fig. S6), we stimulated half of the neural population for each of the two memories. We therefore also showed that drift happens when only a subset of neuron was simulated.

The following is the authors’ response to the original reviews.

**Reviewer #1 (Public Review):**
Current experimental work reveals that brain areas implicated in episodic and spatial memory have a dynamic code, in which activity r imulated networks for epresenting familiar events/locations changes over time. This paper shows that such reconfiguration is consistent with underlying changes in the excitability of cells in the population, which ties these observations to a physiological mechanism.Delamare et al. use a recurrent network model to consider the hypothesis that slow fluctuations in intrinsic excitability, together with spontaneous reactivations of ensembles, may cause the structure of the ensemble to change, consistent with the phenomenon of representational drift. The paper focuses on three main findings from their model: (1) fluctuations in intrinsic excitability lead to drift, (2) this drift has a temporal structure, and (3) a readout neuron can track the drift and continue to decode the memory. This paper is relevant and timely, and the work addresses questions of both a potential mechanism (fluctuations in intrinsic excitability) and purpose (time-stamping memories) of drift.The model used in this study consists of a pool of 50 all-to-all recurrently connected excitatory neurons with weights changing according to a Hebbian rule. All neurons receive the same input during stimulation, as well as global inhibition. The population has heterogeneous excitability, and each neuron's excitability is constant over time apart from a transient increase on a single day. The neurons are divided into ensembles of 10 neurons each, and on each day, a different ensemble receives a transient increase in the excitability of each of its neurons, with each neuron experiencing the same amplitude of increase. Each day for four days, repetitions of a binary stimulus pulse are applied to every neuron.The modeling choices focus in on the parameter of interest-the excitability-and other details are generally kept as straightforward as possible. That said, I wonder if certain aspects may be overly simple. The extent of the work already performed, however, does serve the intended purpose, and so I think it would be sufficient for the authors to comment on these choices rather than to take more space in this paper to actually implement these choices. What might happen were more complex modeling choices made? What is the justification for the choices that are made in the present work?The two specific modeling choices I question are (1) the excitability dynamics and (2) the input stimulus. The ensemble-wide synchronous and constant-amplitude excitability increase, followed by a return to baseline, seems to be a very simplified picture of the dynamics of intrinsic excitability. At the very least, justification for this simplified picture would benefit the reader, and I would be interested in the authors' speculation about how a more complex and biologically realistic dynamics model might impact the drift in their network model. Similarly, the input stimulus being binary means that, on the singleneuron level, the only type of drift that can occur is a sort of drop-in/drop-out drift; this choice excludes the possibility of a neuron maintaining significant tuning to a stimulus but changing its preferred value. How would the use of a continuous input variable influence the results.

(1) In our model, neurons tend to compete for allocation to the memory ensemble: neurons with higher excitability tend to be preferentially allocated and neurons with lower excitability do not respond to the stimulus. Because relative, but not absolute excitability biases this competition, we suggest that the exact distribution of excitability would not impact the results qualitatively. On the other hand, the results might vary if excitability was considered dependent on the activity of the neurons as previously reported experimentally (Cai 2016, Rachid 2016, Pignatelli 2019). An increase in excitability following neural activity might induce higher correlation among ensembles on consecutive days, decreasing the drift.

(2) We thank the reviewer for this very good point. Indeed, two recent studies (Geva 2023 , Khatib 2023) have highlighted distinct mechanisms for a drift of the mean firing rate and the tuning curve. We extended the last part of the discussion to include this point: “Finally, we intended to model drift in the firing rates, as opposed to a drift in the turning curve of the neurons. Recent studies suggest that drifts in the mean firing rate and tuning curve arise from two different mechanisms [33, 34]. Experience drives a drift in neurons turning curve while the passage of time drives a drift in neurons firing rate. In this sense, our study is consistent with these findings by providing a possible mechanism for a drift in the mean firing rates of the neurons driven a dynamical excitability. Our work suggests that drift can depend on any experience having an impact on excitability dynamics such as exercise as previously shown experimentally [9, 35] but also neurogenesis [9, 31, 36], sleep [37] or increase in dopamine level [38]”

Result (1): Fluctuations in intrinsic excitability induce driftThe two choices highlighted above appear to lead to representations that never recruit the neurons in the population with the lowest baseline excitability (Figure 1b: it appears that only 10 neurons ever show high firing rates) and produce networks with very strong bidirectional coupling between this subset of neurons and weak coupling elsewhere (Figure 1d). This low recruitment rate need may not necessarily be problematic, but it stands out as a point that should at least be commented on. The fact that only 10 neurons (20% of the population) are ever recruited in a representation also raises the question of what would happen if the model were scaled up to include more neurons.

This is a very good point. To test how the model depends on the network size, we plotted the drift index against the size of the ensemble. With this current implementation, we did not observe a significant correlation between the drift rate and size of the initial ensemble (Figure S2).

**Author response image 1. sa2fig1:** The rate of the drift does not depend on the size of the engram. Drift rate against the size of the original engram. Each dot shows one simulation (Methods). n = 100 simulations.

Result (2): The observed drift has a temporal structureThe authors then demonstrate that the drift has a temporal structure (i.e., that activity is informative about the day on which it occurs), with methods inspired by Rubin et al. (2015). Rubin et al. (2015) compare single-trial activity patterns on a given session with full-session activity patterns from each session. In contrast, Delamare et al. here compare full-session patterns with baseline excitability (E = 0) patterns. This point of difference should be motivated. What does a comparison to this baseline excitability activity pattern tell us? The ordinal decoder, which decodes the session order, gives very interesting results: that an intermediate amplitude E of excitability increase maximizes this decoder's performance. This point is also discussed well by the authors. As a potential point of further exploration, the use of baseline excitability patterns in the day decoder had me wondering how the ordinal decoder would perform with these baseline patterns.

This is a good point. Here, we aimed at dissociating the role of excitability from the one of the recurrent currents. We introduced a time decoder that compares the pattern with baseline excitability (E = 0), in order to test whether the temporal information was encoded in the ensemble i.e. in the recurrent weights. By contrast, because the neural activity is by construction biased towards excitability, a time decoder performed on the full session would work in a trivial way.

Result (3): A readout neuron can track driftThe authors conclude their work by connecting a readout neuron to the population with plastic weights evolving via a Hebbian rule. They show that this neuron can track the drifting ensemble by adjusting its weights. These results are shown very neatly and effectively and corroborate existing work that they cite very clearly.Overall, this paper is well-organized, offers a straightforward model of dynamic intrinsic excitability, and provides relevant results with appropriate interpretations. The methods could benefit from more justification of certain modeling choices, and/or an exploration (either speculative or viaimplementation) of what would happen with more complex choices. This modeling work paves the way for further explorations of how intrinsic excitability fluctuations influence drifting representations.
**Reviewer #2 (Public Review):**
In this computational study, Delamare et al identify slow neuronal excitability as one mechanism underlying representational drift in recurrent neuronal networks and that the drift is informative about the temporal structure of the memory and when it has been formed. The manuscript is very well written and addresses a timely as well as important topic in current neuroscience namely the mechanisms that may underlie representational drift.The study is based on an all-to-all recurrent neuronal network with synapses following Hebbian plasticity rules. On the first day, a cue-related representation is formed in that network and on the next 3 days it is recalled spontaneously or due to a memory-related cue. One major observation is that representational drift emerges day-by-day based on intrinsic excitability with the most excitable cells showing highest probability to replace previously active members of the assembly. By using a daydecoder, the authors state that they can infer the order at which the reactivation of cell assemblies happened but only if the excitability state was not too high. By applying a read-out neuron, the authors observed that this cell can track the drifting ensemble which is based on changes of the synaptic weights across time. The only few questions which emerged and could be addressed either theoretically or in the discussion are as follows:1. Would the similar results be obtained if not all-to-all recurrent connections would have been molded but more realistic connectivity profiles such as estimated for CA1 and CA3?

This is a very interesting point. We performed further simulations to show that the results are not dependent on the exact structure of the network. In particular, we show that all-to-all connectivity is not required to observe a drift of the ensemble. We found similar results when the recurrent weights matrix was made sparse (Fig. S4a-c, Methods). Similarly to all-to-all connectivity, we found that the ensemble is informative about its temporal history (Fig. S4d) and that an output neuron can decode the ensemble continuously (Fig. S4e).

**Author response image 2. sa2fig2:** Sparse recurrent connectivity shows similar drifting behavior as all-to-all connectivity. The same simulation protocol as Fig. 1 was used while the recurrent weights matrix was made 50% sparse (Methods). (a) Firing rates of the neurons across time. The red traces correspond to neurons belonging to the first assembly, namely that have a firing rate higher than the active threshold after the first stimulation. The black bars show the stimulation and the dashed line shows the active threshold. (b) Recurrent weights matrices after each of the four stimuli show the drifting assembly. (c) Correlation of the patterns of activity between the first day and every other days. (d) Student's test t-value of the ordinal time decoder, for the real (red) and shuffled (orange) data and for different amplitudes of excitability E. (e) Center of mass of the distribution of the output weights (Methods) across days. (c-e) Data are shown as mean ± s.e.m. for n = 10 simulations.

1. How does the number of excited cells that could potentially contribute to an engram influence the representational drift and the decoding quality?

This is indeed a very good question. We did not observe a significant correlation between the drift rate and size of the initial ensemble (Fig. S2).

**Author response image 3. sa2fig3:** The rate of the drift does not depend on the size of the engram. Drift rate against the size of the original engram. Each dot shows one simulation (Methods). n = 100 simulations.

1. How does the rate of the drift influence the quality of readout from the readout-out neuron?

We thank the reviewer for this interesting question. We introduced a measure of the “read-out quality” and plotted this value against the rate of the drift. We found a small correlation between the two quantities. Indeed, the read-out quality decreases with the rate of the drift.

**Author response image 4. sa2fig4:** The quality of the read-out decreases with the rate of the drift. Read-out quality computed on the firing rate of the output neuron against the rate of the drift (Methods). Each dot shows one simulation. n = 100 simulations.

**Reviewer #3 (Public Review):**
The authors explore an important question concerning the underlying mechanism of representational drift, which despite intense recent interest remains obscure. The paper explores the intriguing hypothesis that drift may reflect changes in the intrinsic excitability of neurons. The authors set out to provide theoretical insight into this potential mechanism.They construct a rate model with all-to-all recurrent connectivity, in which recurrent synapses are governed by a standard Hebbian plasticity rule. This network receives a global input, constant across all neurons, which can be varied with time. Each neuron also is driven by an "intrinsic excitability" bias term, which does vary across cells. The authors study how activity in the network evolves as this intrinsic excitability term is changed.They find that after initial stimulation of the network, those neurons where the excitability term is set high become more strongly connected and are in turn more responsive to the input. Each day the subset of neurons with high intrinsic excitability is changed, and the network's recurrent synaptic connectivity and responsiveness gradually shift, such that the new high intrinsic excitability subset becomes both more strongly activated by the global input and also more strongly recurrently connected. These changes result in drift, reflected by a gradual decrease across time in the correlation of the neuronal population vector response to the stimulus.The authors are able to build a classifier that decodes the "day" (i.e. which subset of neurons had high intrinsic excitability) with perfect accuracy. This is despite the fact that the excitability bias during decoding is set to 0 for all neurons, and so the decoder is really detecting those neurons with strong recurrent connectivity, and in turn strong responses to the input. The authors show that it is also possible to decode the order in which different subsets of neurons were given high intrinsic excitability on previous "days". This second result depends on the extent by which intrinsic excitability was increased: if the increase in intrinsic excitability was either too high or too low, it was not possible to read out any information about past ordering of excitability changes.Finally, using another Hebbian learning rule, the authors show that an output neuron, whose activity is a weighted sum of the activity of all neurons in the network, is able to read out the activity of the network. What this means specifically, is that although the set of neurons most active in the network changes, the output neuron always maintains a higher firing rate than a neuron with randomly shuffled synaptic weights, because the output neuron continuously updates its weights to sample from the highly active population at any given moment. Thus, the output neuron can readout a stable memory despite drift.Strengths:The authors are clear in their description of the network they construct and in their results. They convincingly show that when they change their "intrinsic excitability term", upon stimulation, the Hebbian synapses in their network gradually evolve, and the combined synaptic connectivity and altered excitability result in drifting patterns of activity in response to an unchanging input (Fig. 1, Fig. 2a). Furthermore, their classification analyses (Fig. 2) show that information is preserved in the network, and their readout neuron successfully tracks the active cells (Fig. 3). Finally, the observation that only a specific range of excitability bias values permits decoding of the temporal structure of the history of intrinsic excitability (Fig. 2f and Figure S1) is interesting, and as the authors point out, not trivial.Weaknesses:1. The way the network is constructed, there is no formal difference between what the authors call "input", Δ(t), and what they call "intrinsic excitability" Ɛ_i(t) (see Equation 3). These are two separate terms that are summed (Eq. 3) to define the rate dynamics of the network. The authors could have switched the names of these terms: Δ(t) could have been considered a global "intrinsic excitability term" that varied with time and Ɛ_i(t) could have been the external input received by each neuron i in the network. In that case, the paper would have considered the consequence of "slow fluctuations of external input" rather than "slow fluctuations of intrinsic excitability", but the results would have been the same. The difference is therefore semantic. The consequence is that this paper is not necessarily about "intrinsic excitability", rather it considers how a Hebbian network responds to changes in excitatory drive, regardless of whether those drives are labeled "input" or "intrinsic excitability".

This is a very good point. We performed further simulations to model “slope-based”, instead of “threshold-based”, changes in excitability (Fig. S5a, Methods). In this new definition of excitability, we changed the slope of the activation function, which is initially sampled from a random distribution. By introducing a varying excitability, we found very similar results than when excitability was varied as the threshold of the activation function (Fig. S5b-d). We also found similarly that the ensemble is informative about its temporal history (Fig. S5e) and that an output neuron can decode the ensemble continuously (Fig. S5f).

**Author response image 5. sa2fig5:** Change of excitability as a variable slope of the input-output function shows similar drifting behavior as considering a change in the threshold. The same simulation protocol as Fig. 1 was used while the excitability changes were modeled as a change in the activation function slope (Methods). (a) Schema showing two different ways of defining excitability, as a threshold (top) or slope (bottom) of the activation function. Each line shows one neuron and darker lines correspond to neurons with increased excitability. (b) Firing rates of the neurons across time. The red traces correspond to neurons belonging to the first assembly, namely that have a firing rate higher than the active threshold after the first stimulation. The black bars show the stimulation and the dashed line shows the active threshold. (c) Recurrent weights matrices after each of the four stimuli show the drifting assembly. (d) Correlation of the patterns of activity between the first day and every other days. (e) Student's test t-value of the ordinal time decoder, for the real (red) and shuffled (orange) data and for different amplitudes of excitability E. (f) Center of mass of the distribution of the output weights (Methods) across days. (d-f) Data are shown as mean ± s.e.m. for n = 10 simulations.

1. Given how the learning rule that defines input to the readout neuron is constructed, it is trivial that this unit responds to the most active neurons in the network, more so than a neuron assigned random weights. What would happen if the network included more than one "memory"? Would it be possible to construct a readout neuron that could classify two distinct patterns? Along these lines, what if there were multiple, distinct stimuli used to drive this network, rather than the global input the authors employ here? Does the system, as constructed, have the capacity to provide two distinct patterns of activity in response to two distinct inputs?

This is an interesting point. In order to model multiple memories, we introduced non-uniform feedforward inputs, defining different “contexts” (Methods). We adapted our model so that two contexts target two random sub-populations in the network. We also introduced a second output neuron to decode the second memory. The simulation protocol was adapted so that each of the two contexts are stimulated every day (Fig. S6a). We found that the network is able to store two ensembles that drift independently (Fig. S6 and S7a). We were also able to decode temporal information from the patterns of activity of both ensembles (Fig. S7b). Finally, both memories could be decoded independently using two output neurons (Fig. S7c and d).

**Author response image 6. sa2fig6:** Two distinct ensembles can be encoded and drift independently. a) and b) Firing rates of the neurons across time. The red traces in panel b) correspond to neurons belonging to the first assembly and the green traces to the second assembly on the first day. They correspond to neurons having a firing rate higher than the active threshold after the first stimulation of each assembly. The black bars show the stimulation and the dashed line shows the active threshold. c) Recurrent weights matrices after each of the eight stimuli showing the drifting of the first (top) and second (bottom) assembly.

**Author response image 7. sa2fig7:** The two ensembles are informative about their temporal history and can be decoded using two output neurons. a) Correlation of the patterns of activity between the first day and every other days, for the first assembly (red) and the second assembly (green). b) Student's test t-value of the ordinal time decoder, for the first (red, left) and second ensemble (green, right) for different amplitudes of excitability E. Shuffled data are shown in orange. c) Center of mass of the distribution of the output weights (Methods) across days for the first (w?ut , red) and second (W20L't , green) ensemble. a-c) Data are shown as mean ± s.e.m. for n = 10 simulations. d) Output neurons firing rate across time for the first ensemble (Yl, top) and the second ensemble (h, bottom). The red and green traces correspond to the real output. The dark blue, light blue and yellow traces correspond to the cases where the output weights were randomly shuffled for every time points after presentation of the first, second and third stimulus, respectively.

Impact:Defining the potential role of changes in intrinsic excitability in drift is fundamental. Thus, this paper represents a potentially important contribution. Unfortunately, given the way the network employed here is constructed, it is difficult to tease apart the specific contribution of changing excitability from changing input. This limits the interpretability and applicability of the results.